# Identifying metabolic features of colorectal cancer liability using Mendelian randomization

Caroline Bull[1,2,3†], Emma Hazelwood[1,2†], Joshua A Bell[1,2], Vanessa Tan[1,2], Andrei-Emil Constantinescu[1,2], Carolina Borges[1,2], Danny Legge[3], Kimberley Burrows[1,2], Jeroen R Huyghe[4], Hermann Brenner[5,6,7], Sergi Castellvi-Bel[8], Andrew T Chan[9,10,11,12,13,14], Sun-Seog Kweon[15,16], Loic Le Marchand[17], Li Li[18], Iona Cheng[19,20], Rish K Pai[21], Jane C Figueiredo[22], Neil Murphy[23], Marc J Gunter[24,25], Nicholas J Timpson[1,2], Emma E Vincent[1,2,3]*

[1]MRC Integrative Epidemiology Unit at the University of Bristol, Bristol, United Kingdom; [2]Population Health Sciences, Bristol Medical School, University of Bristol, Bristol, United Kingdom; [3]Translational Health Sciences, Bristol Medical School, University of Bristol, Bristol, United Kingdom; [4]Public Health Sciences Division, Fred Hutchinson Cancer Center, Seattle, United States; [5]Division of Clinical Epidemiology and Aging Research, German Cancer Research Center (DKFZ), Heidelberg, Germany; [6]Division of Preventive Oncology, German Cancer Research Center (DKFZ) and National Center for Tumor Diseases (NCT), Heidelberg, Germany; [7]German Cancer Consortium (DKTK), German Cancer Research Center (DKFZ), Heidelberg, Germany; [8]Gastroenterology Department, Hospital Clínic, Institut d'Investigacions Biomèdiques August Pi i Sunyer (IDIBAPS), Centro de Investigación Biomédica en Red de Enfermedades Hepáticas y Digestivas (CIBEREHD), University of Barcelona, Barcelona, Spain; [9]Division of Gastroenterology, Massachusetts General Hospital and Harvard Medical School, Boston, United States; [10]Channing Division of Network Medicine, Brigham and Women's Hospital and Harvard Medical School, Boston, United States; [11]Clinical and Translational Epidemiology Unit, Massachusetts General Hospital and Harvard Medical School, Boston, United States; [12]Broad Institute of Harvard and MIT, Cambridge, United States; [13]Department of Epidemiology, Harvard T.H. Chan School of Public Health, Harvard University, Boston, United States; [14]Department of Immunology and Infectious Diseases, Harvard T.H. Chan School of Public Health, Harvard University, Boston, United States; [15]Department of Preventive Medicine, Chonnam National University Medical School, Gwangju, Republic of Korea; [16]Jeonnam Regional Cancer Center, Chonnam National University Hwasun Hospital, Hwasun, Republic of Korea; [17]University of Hawaii Cancer Center, Honolulu, United States; [18]Department of Family Medicine, University of Virginia, Charlottesville, United States; [19]Department of Epidemiology and Biostatistics, University of California, San Francisco, San Francisco, United States; [20]University of California, San Francisco Helen Diller Family Comprehensive Cancer Center, San Francisco, San Francisco, United States; [21]Department of Pathology and Laboratory Medicine, Mayo Clinic, Scottsdale, United States; [22]Department of Medicine, Samuel Oschin Comprehensive Cancer Institute, Cedars-Sinai Medical Center, Los Angeles, United States; [23]Nutrition and Metabolism Branch, International Agency for Research on Cancer, Lyon, France; [24]Nutrition and Metabolism Branch, International Agency for Research on Cancer, Lyon, France; [25]Department of Epidemiology and

*For correspondence: emma.vincent@bristol.ac.uk

†These authors contributed equally to this work

Competing interest: The authors declare that no competing interests exist.

Biostatistics, School of Public Health, Imperial College London, London, United Kingdom

## Abstract

**Background:** Recognizing the early signs of cancer risk is vital for informing prevention, early detection, and survival.

**Methods:** To investigate whether changes in circulating metabolites characterize the early stages of colorectal cancer (CRC) development, we examined the associations between a genetic risk score (GRS) associated with CRC liability (72 single-nucleotide polymorphisms) and 231 circulating metabolites measured by nuclear magnetic resonance spectroscopy in the Avon Longitudinal Study of Parents and Children (N = 6221). Linear regression models were applied to examine the associations between genetic liability to CRC and circulating metabolites measured in the same individuals at age 8 y, 16 y, 18 y, and 25 y.

**Results:** The GRS for CRC was associated with up to 28% of the circulating metabolites at FDR-P < 0.05 across all time points, particularly with higher fatty acids and very-low- and low-density lipoprotein subclass lipids. Two-sample reverse Mendelian randomization (MR) analyses investigating CRC liability (52,775 cases, 45,940 controls) and metabolites measured in a random subset of UK Biobank participants (N = 118,466, median age 58 y) revealed broadly consistent effect estimates with the GRS analysis. In conventional (forward) MR analyses, genetically predicted polyunsaturated fatty acid concentrations were most strongly associated with higher CRC risk.

**Conclusions:** These analyses suggest that higher genetic liability to CRC can cause early alterations in systemic metabolism and suggest that fatty acids may play an important role in CRC development.

**Funding:** This work was supported by the Elizabeth Blackwell Institute for Health Research, University of Bristol, the Wellcome Trust, the Medical Research Council, Diabetes UK, the University of Bristol NIHR Biomedical Research Centre, and Cancer Research UK. The funders had no role in study design, data collection and analysis, decision to publish, or preparation of the manuscript. This work used the computational facilities of the Advanced Computing Research Centre, University of Bristol - http://www.bristol.ac.uk/acrc/.

## eLife assessment

The manuscript by Bull et al. provides **valuable** information on the relationship between metabolic features, in particular different lipoproteins and fatty acids, and colorectal cancer. They use **solid** methods and combine different data sources to analyze forward and reverse Mendelian randomizations that support their claims.

## Introduction

Colorectal cancer (CRC) is the third most frequently diagnosed cancer worldwide and the fourth most common cause of death from cancer (*Ferlay et al., 2015*; *Clinton et al., 2018*). There is a genetic component to risk of the disease, which is thought to explain up to 35% of variability in CRC risk (*Huyghe et al., 2019*; *Czene et al., 2002*; *Lichtenstein et al., 2000*). In addition, modifiable lifestyle factors, including obesity, consumption of processed meat, and alcohol, are thought to increase CRC risk (*Clinton et al., 2018*; *Händel et al., 2020*; *McNabb et al., 2020*; *Lauby-Secretan et al., 2016*; *Gui et al., 2023*). However, the underlying biological pathways remain unclear, which limits targeted prevention strategies. While CRC has higher mortality rates when diagnosed at later stages, early-stage CRC or precancerous lesions are largely treatable, meaning CRC screening programmes have the potential to be highly effective (*Meester et al., 2020*; *Cardoso et al., 2021*). Due to the lack of known predictive biomarkers for CRC, wide-scale screening (if implemented at all) is expensive and

**eLife digest** Colorectal cancer, or bowel cancer, is the fourth most common cause of death from cancer worldwide. Understanding how the cancer develops and recognizing early signs is essential, as people who receive treatment early on have higher survival rates.

One way to boost early detection and disease survival rates is through identifying early colorectal cancer biomarkers. For example, metabolites produced when cells process nutrients have been shown to play a role in the development of colon cancer. Certain metabolites could therefore serve as biomarkers, which can be detected in routine blood tests. But first, scientists need to identify the exact metabolic processes involved in cancer development.

Bull, Hazelwood et al. show that fat metabolites during early adulthood may help predict colorectal cancer risk. In the experiments, the team assessed the link between an individual's genetic risk for developing colorectal cancer and metabolites in their blood. By looking at data from over 6,000 individuals living in the UK, followed from early life into adulthood, they found higher fatty acid and low-density lipoprotein levels in young adults at risk of colorectal cancer. However, the results could not be replicated in a separate cohort study of middle-aged adults. Bull, Hazelwood et al. noted that many individuals in this older age group use fat-targeting drugs called statins, which may have obscured this connection.

The study of Bull, Hazelwood et al. shows that colorectal cancer risk indicators may be present from adolescence to around 40 years, before most individuals are diagnosed. The results suggest this may be a window for early detection and preventive interventions. It also highlights that differences in fat metabolism, possibly linked to genetic differences, may underlie colorectal cancer risk. More studies are needed to better understand how and whether interventions targeting fat levels may help prevent colorectal cancer development.

often targeted crudely by age range. Identifying biomarkers predictive of CRC, or with causal roles in disease development, is therefore vital.

One potential source of biomarkers for CRC risk is the circulating metabolome, which offers a dynamic insight into cellular processes and disease states. It is increasingly clear from mechanistic studies that both systemic and intracellular tumour metabolism play an important role in CRC development and progression (*Qiu et al., 2009*; *Ward and Thompson, 2012*). Interestingly, several major risk factors for CRC are known to have profound effects on metabolism (*Rattray et al., 2017*). For instance, obesity has been shown via conventional observational and Mendelian randomization (MR) analyses to strongly alter circulating metabolite levels (*Gui et al., 2023*; *Singla et al., 2010*; *Papandreou et al., 2021*; *Ahmad et al., 2022*). This suggests that the circulating metabolome may play a mediating role in the relationship between at least some common risk factors, such as obesity, and CRC – or at least might be a useful biomarker for disease or intermediates thereof. In particular, previous work has highlighted polyunsaturated fatty acids (PUFAs) as potentially having a role in CRC development. The term PUFA includes omega-3 and -6 fatty acids. Recent MR work has highlighted a possible link between PUFAs, in particular omega-6 PUFAs, and CRC risk (*Haycock et al., 2023*). Further investigating the relationship between CRC and circulating metabolites may therefore provide powerful insights into the causal pathways underlying disease risk or alternatively may be valuable in prediction and early diagnosis.

MR is a genetic epidemiological approach used to evaluate causal relationships between traits (*Yarmolinsky et al., 2018*; *Smith and Ebrahim, 2003*). This method uses genetic variation as a proxy measure for traits in an instrumental variable framework to assess the causal relevance of the traits in disease development. As germline genetic variants are theoretically randomized between generations and fixed at conception, this approach should be less prone to bias and confounding than conventional analyses undertaken in an observational context. Conventionally, MR is used to investigate the effect of an exposure on a disease outcome. In reverse MR, genetic instruments proxy the association between liability to a disease and other traits (*Holmes and Davey Smith, 2019*). This approach can identify the biomarkers which cause the disease, are predictive for the disease, or have diagnostic potential (*Holmes and Davey Smith, 2019*). Given the suspected importance of the circulating metabolome in CRC development, employing both reverse MR and conventional forward MR

for metabolites in the same study may be an efficient approach for revealing causal and predictive biomarkers for CRC. Although previous observational studies have investigated associations between the circulating metabolome and CRC risk, these studies may have been influenced by confounding bias, which should be less relevant to MR analyses (*Gao et al., 2023*; *Rattner et al., 2022*; *Leichtle et al., 2012*; *Ritchie et al., 2010*; *Nishiumi et al., 2012*; *Ma et al., 2012*; *Bertini et al., 2012*; *Zhang et al., 2014*; *Farshidfar et al., 2016*; *Farshidfar et al., 2012*). Additionally, these studies focussed on adults, who commonly take medications which may confound metabolite associations, further complicating interpretations.

Here, we applied a reverse MR framework to identify circulating metabolites which are associated with CRC liability across different stages of the early life course (spanning childhood to young adulthood, when use of medications and CRC are both rare) using data from a birth cohort study. We then attempted to replicate these results using reverse two-sample MR in an independent cohort of middle-aged adults (UK Biobank). We then performed conventional 'forward' MR of metabolites onto CRC risk using large-scale cancer consortia data to identify the metabolites which may have a causal role in CRC development.

## Methods
### Study populations

This study uses data from two cohort studies: the Avon Longitudinal Study of Parents and Children (ALSPAC) offspring (generation 1) cohort (individual-level data) and the UK Biobank cohort (summary-level data); plus summary-level data from a genome-wide association study (GWAS) meta-analysis of CRC comprising the Genetics and Epidemiology of Colorectal Cancer Consortium (GECCO), Colorectal Transdisciplinary Study (CORECT), and Colon Cancer Family Registry (CCFR).

ALSPAC is a population-based birth cohort study in which 14,541 pregnant women with an expected delivery date between 1 April 1991 and 31 December 1992 were recruited from the former Avon County of southwest England (*Boyd et al., 2013*). Since then, 13,988 offspring alive at 1 y have been followed repeatedly with questionnaire- and clinic-based assessments (*Fraser et al., 2013*; *Northstone et al., 2019*). Sufficient information was available on 6221 of these individuals to be included in our analysis, as metabolomics was not performed for all individuals in the ALSPAC study. Study data were collected and managed using REDCap (Research Electronic Data Capture) electronic data capture tools hosted at the University of Bristol (*Harris et al., 2009*) REDCap is a secure, web-based software platform designed to support data capture for research studies. Offspring genotype was assessed using the Illumina HumanHap550 quad chip platform. Quality control measures included exclusion of participants with sex mismatch, minimal or excessive heterozygosity, disproportionately missing data, insufficient sample replication, cryptic relatedness, and non-European ancestry. Imputation was performed using the Haplotype Reference Consortium (HRC) panel. Offspring were considered for the current analyses if they had no older siblings in ALSPAC (203 excluded) and were of white ethnicity (based on reports by parents, 604 excluded) to reduce the potential for confounding by genotype. The study website contains details of all available data through a fully searchable data dictionary and variable search tool (http://www.bristol.ac.uk/alspac/researchers/our-data/).

UK Biobank is a population-based cohort study based in 22 centres across the UK (*Sudlow et al., 2015*). The cohort is made up of around 500,000 adults aged 40–80 years old, who were enrolled between 2006 and 2010. Genotyping data is available for 488,377 participants (*Bycroft et al., 2018*). Participants were genotyped using one of two arrays – either the Applied Biosystems UK BiLEVE Axiom Array by Affymetrix (now part of Thermo Fisher Scientific) or the closely related Applied Biosystems UK Biobank Axiom Array. Approaches based on principal component analysis (PCA) were used to account for population structure. Individuals were excluded if reported sex differed from inferred sex based on genotyping data; if they had sex chromosome karyotypes which were not XX or XY; if they were outliers in terms of heterozygosity and missing rates; or if they had high relatedness to another participant. Multiallelic SNPs or those with a minor allele frequency of below 1% were removed. Imputation was performed using the UK10K haplotype and HRC reference panels.

The GWAS meta-analysis for CRC included up to 52,775 cases and 45,940 controls (*Huyghe et al., 2019*; *Huyghe et al., 2021*). This sample excluded cases and controls from UK Biobank to avoid potential bias due to sample overlap which may be problematic in MR analyses (*Burgess et al., 2016*).

Cases were diagnosed by a physician and recorded overall and by site (colon, 28,736 cases; proximal colon, 14,416 cases; distal colon, 12,879 cases; and rectal, 14,150 cases). Colon cancer included proximal colon (any primary tumour arising in the caecum, ascending colon, hepatic flexure, or transverse colon), distal colon (any primary tumour arising in the pleenic flexure, descending colon, or sigmoid colon), and colon cases with unspecified site. Rectal cancer included any primary tumour arising in the rectum or rectosigmoid junction (*Huyghe et al., 2019*). Approximately 92% of participants in the overall CRC GWAS were white European (~8% were East Asian). All participants included in site-specific CRC analyses were of European ancestry. Imputation was performed using the Michigan imputation server and HRC r1.0 reference panel. Regression models were further adjusted for age, sex, genotyping platform, and genomic principal components as described previously (*Huyghe et al., 2019*).

## Assessment of CRC genetic liability

Genetic liability to CRC was based on single-nucleotide polymorphisms (SNPs) associated with CRC case status at genome-wide significance ($p<5 \times 10^{-8}$). A total of 108 independent SNPs reported by two major GWAS meta-analyses were eligible for inclusion in a CRC genetic risk score (GRS) (*Huyghe et al., 2019*; *Law et al., 2019*). The set of SNPs was filtered, excluding 36 SNPs that were in linkage disequilibrium based on $R^2 > 0.001$ using the TwoSampleMR package (SNPs with the lowest p-values were retained) (*Hemani et al., 2018*). This left 72 SNPs independently associated with CRC (*Supplementary file 1a*), 65 of which were available in imputed ALSPAC genotype data post quality control. As GWAS of site-specific CRC have identified marked heterogeneity (*Huyghe et al., 2021*), GRS describing site-specific CRCs were constructed for sensitivity analyses using the same process outlined above. The GRS for colon cancer, rectal cancer, proximal colon cancer, and distal colon cancer were comprised of 38, 25, 20, and 24 variants, respectively (*Supplementary file 1a*). For overall CRC and site-specific CRC analyses, sensitivity analyses excluding any SNPs in the FADS cluster (i.e. within the gene regions of *FADS1*, *FADS2*, or *FADS3*) (*Supplementary file 1a*) were performed given a likely role for these SNPs in influencing circulating metabolite levels directly, in particular via lipid metabolism (i.e. not primarily due to CRC) (*Lu et al., 2019*; *Zaytseva et al., 2018*; *ClinicalTrials.gov, 2022*; *Fhu and Ali, 2020*; *Chen et al., 2020*; *Kathiresan et al., 2009*; *Tanaka et al., 2009*).

## Assessment of circulating metabolites

Circulating metabolite measures were drawn from ALSPAC and UK Biobank using the same targeted metabolomics platform. In ALSPAC, participants provided non-fasting blood samples during a clinic visit while aged approximately 8 y, and fasting blood samples from clinic visits while aged approximately 16 y, 18 y, and 25 y. Proton nuclear magnetic resonance ($^1$H-NMR) spectroscopy was performed on ethylenediaminetetraacetic acid (EDTA) plasma (stored at or below –70°C pre-processing) to quantify a maximum of 231 metabolites (*Würtz et al., 2017*). Quantified metabolites included the cholesterol and triglyceride content of lipoprotein particles; the concentrations and diameter/size of these particles; apolipoprotein B and apolipoprotein A-1 concentrations; as well as fatty acids and their ratios to total fatty acid concentration, branched chain and aromatic amino acids, glucose and pre-glycaemic factors including lactate and citrate, fluid balance factors including albumin and creatinine, and the inflammatory marker glycoprotein acetyls (GlycA). This metabolomics platform has limited coverage of fatty acids. In UK Biobank, EDTA plasma samples from 117,121 participants, a random subset of the original ~500,000 who provided samples at assessment centres between 2006 and 2013, were analysed between 2019 and 2020 for levels of 249 metabolic traits (168 concentrations plus 81 ratios) using the same high-throughput $^1$H-NMR platform. Data pre-processing and QC steps are described previously (*Würtz et al., 2017*; *Julkunen et al., 2021*; *Bycroft et al., 2018*). To allow comparability between MR and GRS estimates, all metabolite measures were standardized and normalized using rank-based inverse normal transformation. For descriptive purposes in ALSPAC, body mass index (BMI) was calculated at each time point as weight (kg) divided by squared height (m$^2$) based on clinic measures of weight to the nearest 0.1 kg using a Tanita scale and height measured in light clothing without shoes to the nearest 0.1 cm using a Harpenden stadiometer.

CRC liability variants were combined into a GRS using PLINK 1.9, specifying the effect (risk raising) allele and coefficient (logOR) with estimates from the CRC GWAS used as external weights (*Huyghe et al., 2019*; *Law et al., 2019*). GRSs were calculated as the number of effect alleles (or dosages if

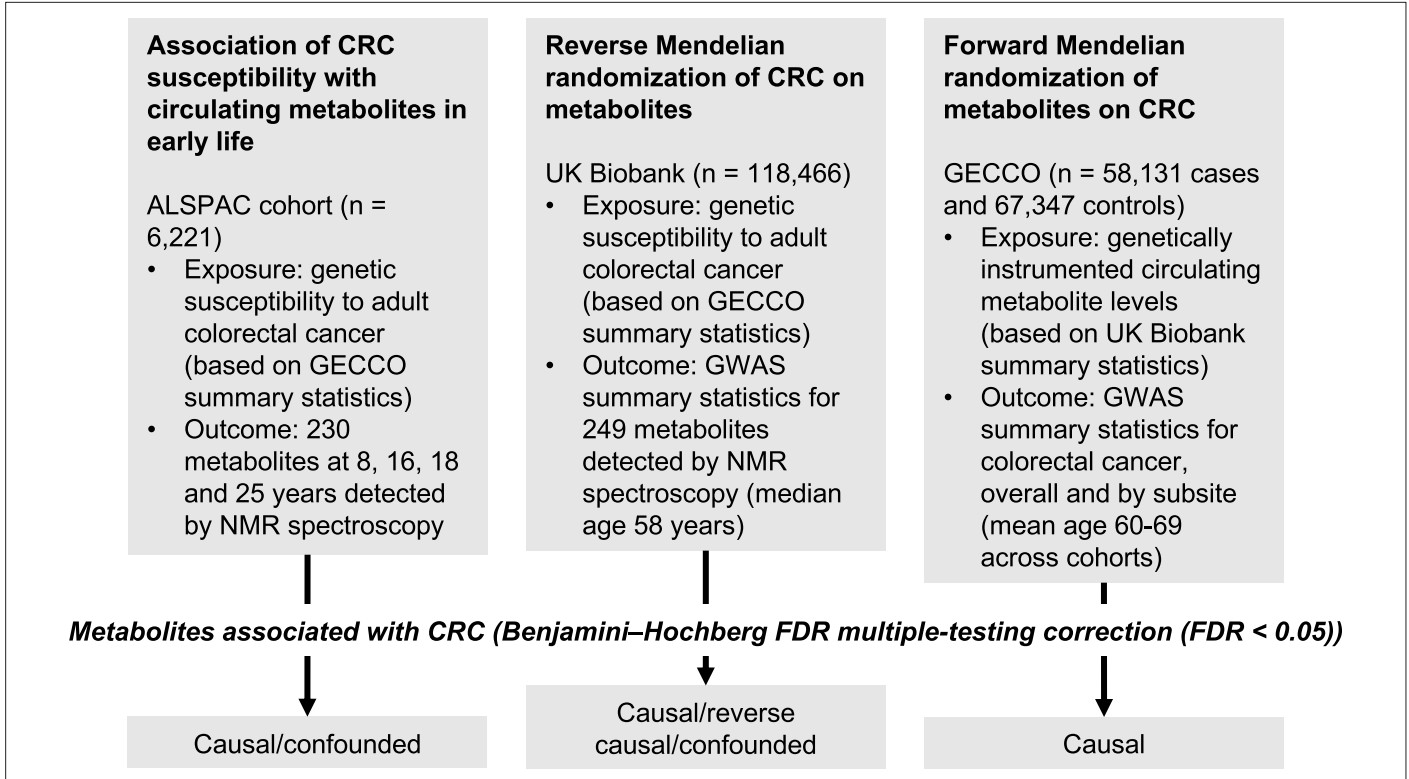

**Figure 1.** Study design. First, linear regression models were used to examine the relationship between genetic susceptibility to adult colorectal cancer (CRC) and circulating metabolites measured in the Avon Longitudinal Study of Parents and Children (ALSPAC) participants at age 8 y, 16 y, 18 y, and 25 y. Next, we performed a reverse Mendelian randomization analysis to identify metabolites influenced by CRC susceptibility in an independent population of adults. Finally, we performed a conventional (forward) Mendelian randomization analysis of circulating metabolites on CRC to identify metabolites causally associated with CRC risk. Consistent evidence across all three methodological approaches was interpreted to indicate a causal role for a given metabolite in CRC aetiology.

imputed) at each SNP (0, 1, or 2) multiplied by its weighting, summing these, and dividing by the total number of SNPs used. Z-scores of GRS variables were calculated to standardize scoring.

## Statistical approach

An overview of the study design is presented in *Figure 1*. To estimate the effect of increased genetic liability to CRC on circulating metabolites, we conducted a GRS analysis in ALSPAC and reverse two-sample MR analyses in UK Biobank. Estimates were interpreted within a 'reverse MR' framework (*Holmes and Davey Smith, 2019*), wherein results are taken to reflect 'metabolic features' of CRC liability which could capture causal or predictive metabolite–disease associations. To clarify the direction of metabolite–CRC associations, we additionally performed conventional 'forward' two-sample MR analyses to estimate the effect of circulating metabolites on CRC risk using large-scale GWAS data on metabolites and CRC.

### Associations of CRC liability with circulating metabolites in early life

Separate linear regression models with robust standard errors were used to estimate coefficients and 95% confidence intervals (95% CIs) for associations of GRSs with each metabolite as a dependent variable measured on the same individuals at age 8 y, 16 y, 18 y, and 25 y, adjusted for sex and age at the time of metabolite assessment. To aid interpretations, estimates were multiplied by 0.693 ($\log_e 2$) to reflect SD-unit differences in metabolites per doubling of genetic liability to CRC (*Burgess and Labrecque, 2018*). The Benjamini–Hochberg method was used to adjust p-values for multiple testing and an adjusted p-value of <0.05 was used as a heuristic for evidence for association given current sample sizes (*Benjamini and Hochberg, 1995*).

## Reverse MR of the effects of CRC liability on circulating metabolites in middle adulthood

'Reverse' MR analyses (*Holmes and Davey Smith, 2019*) were conducted using UK Biobank for outcome datasets in two-sample MR to examine the effect of CRC liability on circulating metabolites. SNP-outcome (metabolite) estimates were obtained from a GWAS of metabolites in UK Biobank (*Clayton et al., 2022*; *Borges et al., 2022a*). Prior to GWAS, all metabolite measures were standardized and normalized using rank-based inverse normal transformation. Genetic association data for metabolites were retrieved using the MRC IEU UK Biobank GWAS pipeline (*Data.bris, 2022*). Full summary statistics are available via the IEU Open GWAS project (*Holmes and Davey Smith, 2019*; *Elsworth et al., 2020*). Up to three statistical methods were used to generate reverse MR estimates of the effect of CRC liability on circulating metabolites using the TwoSampleMR package (*Hemani et al., 2016*): random-effects inverse variance weighted (IVW), weighted-median, and weighted-mode, which each make differing assumptions about directional pleiotropy and SNP heterogeneity (*Bowden et al., 2016*; *Hartwig et al., 2017*). The IVW MR model will produce biased effect estimates in the presence of horizontal pleiotropy, that is, where one or more genetic variant(s) included in the instrument affect the outcome by a pathway other than through the exposure. In the weighted median model, each genetic variant is weighted according to its distance from the median effect of all genetic variants. Thus, the weighted median model will provide an unbiased estimate when at least 50% of the information in an instrument comes from genetic variants that are not horizontally pleiotropic. The weighted mode model uses a similar approach but weights genetic instruments according to the mean effect. In this model, over 50% of the weight of the genetic instrument can be contributed to by genetic variants which are horizontally pleiotropic, but the most common amount of pleiotropy must be zero (known as the Zero Modal Pleiotropy Assumption [ZEMPA]) (*Hartwig et al., 2017*). As above, estimates were multiplied by 0.693 ($\log_e 2$) to reflect SD-unit differences in metabolites per doubling of genetic liability to CRC (*Burgess and Labrecque, 2018*).

### Forward MR of the effects of metabolites on CRC

Forward MR analyses were conducted using summary statistics from UK Biobank for the same NMR-measured metabolites (SNP-exposure) and from GECCO/CORECT/CCFR as outlined above (SNP-outcome). We identified SNPs that were independently associated ($R^2 < 0.001$ and $p < 5 \times 10^{-8}$) with metabolites from a GWAS of 249 metabolites in UK Biobank described above. As before, we used up to three statistical methods to generate MR estimates of the effect of circulating metabolites on CRC risk (overall and site-specific): random-effects IVW, weighted median, and weighted mode. The Benjamini–Hochberg method was used to adjust p-values for multiple testing and an adjusted p-value of <0.05 was used as a heuristic for nominal evidence for a causal effect (*Benjamini and Hochberg, 1995*). MR outputs are beta coefficients representing the logOR for CRC per SD higher metabolite, exponentiated to reflect the OR for CRC per SD metabolite.

MR analyses were performed in R version 4.0.3 (*R Development Core Team, 2021*) and GRS analyses in Stata 16.1 (StataCorp, College Station, TX). The ggforestplot R package was used to generate results visualizations (*Scheinin et al., 2022*).

## Results

### Associations of CRC liability with circulating metabolites in early life

At the time the ALSPAC blood samples were taken, the mean age of participants was 7.5 y (N = 4767), 15.5 y (N = 2930), 17.8 y (N = 2613), and 24.5 y (N = 2559) for the childhood, early adolescence, late adolescence, and young adulthood time points, respectively. The proportion of participants which were male were 50.5, 47.4, 44.5, and 39.1% and mean BMI was 16.2, 21.4, 22.7, and 24.8 kg/m$^2$ for each time point, respectively. The socio-demographic profile of ALSPAC offspring participants has been reported previously (*Boyd et al., 2013*). The mean and standard deviation (SD) values for metabolites on each measurement occasion in ALSPAC are shown in *Supplementary file 1b*.

In the GRS analysis, there was no strong evidence of association of CRC liability with metabolites at age 8 y (*Supplementary file 1c*). At age 16 y, there was evidence for association with several lipid traits including higher cholesteryl esters to total lipids ratio in large low-density lipoprotein (LDL) (SD change per doubling CRC liability = 0.06, 95% CI = 0.02–0.10) and higher cholesterol in very

small very low-density lipoprotein (VLDL) (SD change per doubling CRC liability = 0.06, 95% CI = 0.03–0.10). There was strong evidence for association with several traits at age 18 y including higher non-high-density lipoprotein (non-HDL) lipids, for example, a 1 doubling CRC liability was associated with higher levels of total cholesterol (SD change = 0.05 95% CI = 0.01–0.09), VLDL-cholesterol (SD change = 0.05, 95% CI = 0.01–0.09), LDL-cholesterol (SD change = 0.06, 95% CI = 0.02–0.09), apolipoproteins (apolipoprotein B [SD change = 0.06, 95% CI = 0.02–0.09]), and fatty acids (omega-3 [SD change = 0.08, 95% CI = 0.04–0.11], docosahexaenoic acid [DHA] [SD change = 0.05, 95% CI = 0.02–0.09]) (*Supplementary file 1c*). *Figure 2* (*Figure 2—figure supplements 1–6*) shows results for all clinically validated metabolites. At age 25 y, there was no strong evidence of association of CRC liability with metabolites. In anatomical site-specific analyses, there was strong evidence for association of liability to colon cancer with omega-3 (SD change = 0.07, 95% CI = 0.03–0.11) and DHA (SD change = 0.07, 95% CI = 0.03–0.10) at age 18 y. There was little evidence for any associations at any other CRC site or age (*Supplementary file 1c*). When SNPs in the FADS cluster gene regions were excluded due to possible horizontal pleiotropy given the role of FADS in lipid metabolism, there was a reduction in strength of evidence for an association of liability to CRC with any metabolite measured, although estimates were in a largely consistent direction with the prior analysis (*Supplementary file 1d*).

## Reverse MR of the effects of CRC liability on circulating metabolites in middle adulthood

All instrument sets from the reverse MR analysis had an F-statistic greater than 10 (minimum F-statistic = 36, median = 40), suggesting that our analyses did not suffer from weak instrument bias (*Supplementary file 1e*). There was little evidence of an association of CRC liability (overall or by anatomical site) on any of the circulating metabolites investigated, including when the SNP in the *FADS* gene region was excluded, based on our pre-determined cut-off of FDR-P < 0.05; however, the direction of effect estimates was largely consistent with those seen in ALSPAC GRS analyses, with higher CRC liability weakly associated with higher non-HDLs, lipoproteins, and fatty acid levels (*Supplementary file 1f and g*). *Figure 3* (*Figure 3—figure supplements 1–3*) shows the results for clinically validated metabolites. In subsite stratified analyses, there was strong evidence for a causal effect of genetic liability to proximal colon cancer on several traits, including total fatty acids (SD change per doubling of liability = 0.02, 95% CI = 0.01–0.04) and omega-6 fatty acids (SD change per doubling of liability = 0.03, 95% CI = 0.01–0.05).

## Forward MR for the effects of metabolites on CRC risk

All instrument sets from the forward MR analysis had an F-statistic greater than 10 (minimum F-statistic = 54, median = 141), suggesting that our analyses were unlikely to suffer from weak instrument bias (*Supplementary file 1h and i*). There was strong evidence for an effect of several fatty acid traits on overall CRC risk, including of omega-3 fatty acids (CRC OR = 1.13, 95% CI = 1.06–1.21), DHA (OR CRC = 1.76, 95% CI = 1.08–1.28), ratio of omega-3 fatty acids to total fatty acids (OR CRC = 1.18, 95% CI = 1.11–1.25), ratio of DHA to total fatty acids (CRC OR = 1.20, 95% CI = 1.10–1.31), and ratio of omega-6 fatty acids to omega-3 fatty acids (CRC OR = 0.86, 95% CI = 0.80–9.13) (*Supplementary file 1j*, *Figure 4*, *Figure 4—figure supplements 1–3*). These estimates were overlapping with variable precision in MR sensitivity models. When SNPs in the *FADS* gene region were excluded, there was little evidence for a causal effect of any metabolite investigated on CRC risk based on the prede-termined FDR-P cut-of off <0.05, although the directions of effect estimates were consistent with previous analyses (*Supplementary file 1k*).

In anatomical subtype-stratified analyses, evidence was the strongest for an effect of fatty acid traits on higher CRC risk, and this appeared specific to the distal colon, for example, omega-3 (distal CRC OR = 1.20, 95% CI = 1.09–1.32), and ratio of DHA to total fatty acids (distal colon OR = 1.29, 95% CI = 1.16–1.43). There was also evidence of a negative effect of ratio of omega-6 to omega-3 fatty acids (distal CRC OR = 0.80, 95% CI = 0.74–0.88) and a positive effect of ratio of omega-3 fatty acids to total fatty acids (distal CRC = 1.24, 95% CI = 1.15–1.35; seen also for proximal CRC OR = 1.15, 95% CI = 1.07–1.23) (*Supplementary file 1j*). These estimates were also directionally consistent in MR sensitivity models.

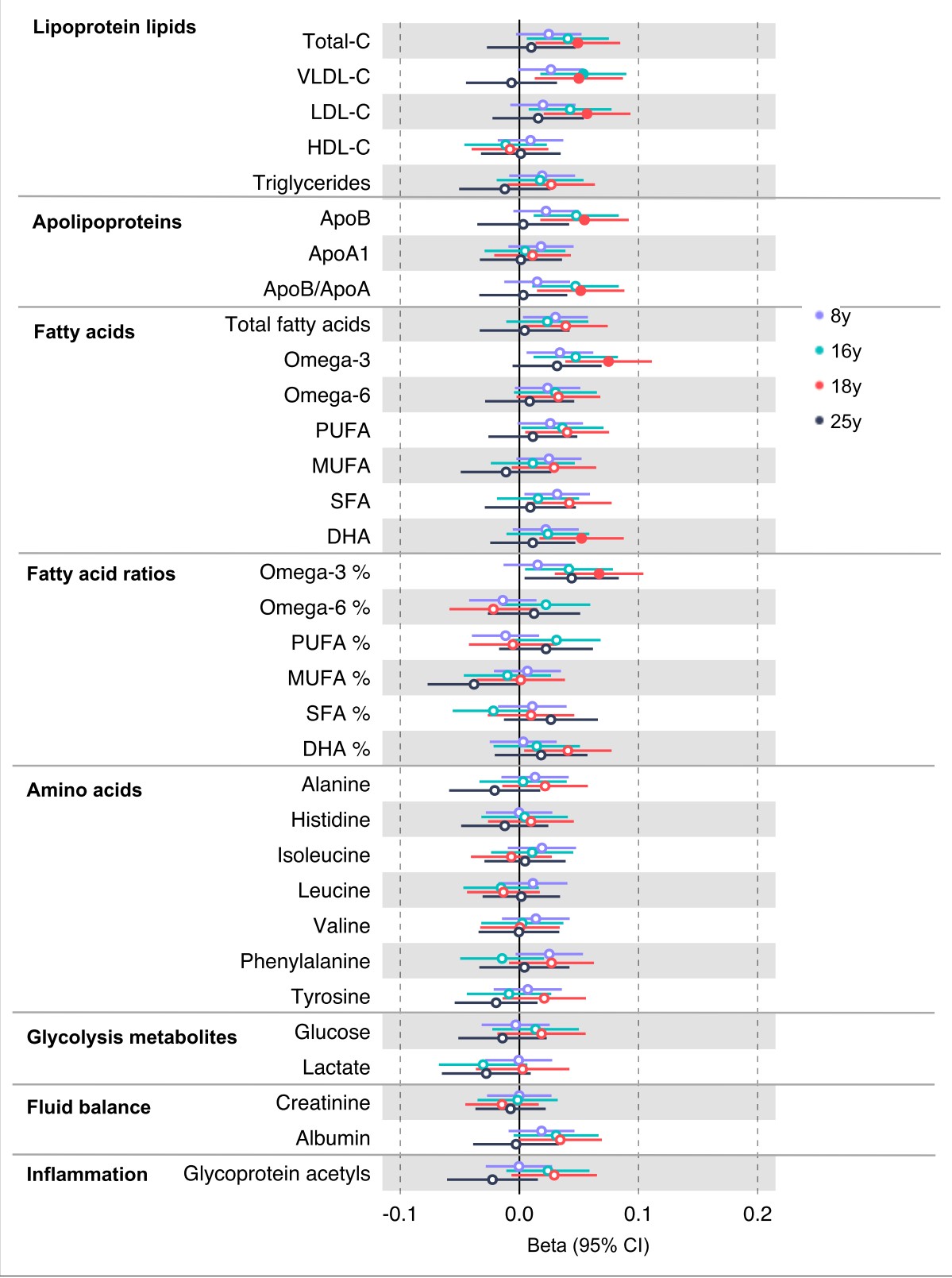

**Figure 2.** Associations of genetic liability to adult colorectal cancer (based on a 72 single-nucleotide polymorphism [SNP] genetic risk score) with clinically validated metabolic traits at different early life stages among the Avon Longitudinal Study of Parents and Children (ALSPAC) offspring (age 8 y [N = 4767], 16 y [N = 2930], 18 y [N = 2613], and 25 y [N = 2559]). Estimates shown are beta coefficients representing the SD difference in metabolic

*Figure 2 continued on next page*

*Figure 2 continued*

trait per doubling of genetic liability to colorectal cancer (purple, 8 y; turquoise, 16 y; red, 18 y; black, 25 y). Filled point estimates are those that pass a Benjamini–Hochberg FDR multiple-testing correction (FDR < 0.05).

The online version of this article includes the following figure supplement(s) for figure 2:

**Figure supplement 1.** Associations of genetic liability to adult colon cancer with clinically validated metabolic traits at different early life stages among the Avon Longitudinal Study of Parents and Children (ALSPAC) offspring (age 8 y, 16 y, 18 y, and 25 y).

**Figure supplement 2.** Associations of genetic liability to proximal colon cancer with clinically validated metabolic traits at different early life stages among the Avon Longitudinal Study of Parents and Children (ALSPAC) offspring (age 8 y, 16 y, 18 y, and 25 y).

**Figure supplement 3.** Associations of genetic liability to distal colon cancer with clinically validated metabolic traits at different early life stages among the Avon Longitudinal Study of Parents and Children (ALSPAC) offspring (age 8 y, 16 y, 18 y, and 25 y).

**Figure supplement 4.** Associations of genetic liability to rectal cancer with clinically validated metabolic traits at different early life stages among the Avon Longitudinal Study of Parents and Children (ALSPAC) offspring (age 8 y, 16 y, 18 y, and 25 y).

**Figure supplement 5.** Associations of genetic liability to adult colorectal cancer (excluding rs174533) with clinically validated metabolic traits at different early life stages among the Avon Longitudinal Study of Parents and Children (ALSPAC) offspring (age 8 y, 16 y, 18 y, and 25 y).

**Figure supplement 6.** Associations of genetic liability to adult colon cancer (excluding rs174535) with clinically validated metabolic traits at different early life stages among the Avon Longitudinal Study of Parents and Children (ALSPAC) offspring (age 8 y, 16 y, 18 y, and 25 y).

## Discussion

Here, we used a reverse MR framework to identify circulating metabolites which are associated with genetic CRC liability across different stages of the early life course and attempted to replicate results in an independent cohort of middle-aged adults. We then performed forward MR to characterize the causal direction of the relationship between metabolites and CRC. Our GRS analysis provided evidence for an association of genetic liability to CRC with higher circulating levels of lipoprotein lipids (including total cholesterol, VLDL-cholesterol, and LDL-cholesterol), apolipoproteins (including apolipoprotein B), and fatty acids (including omega-3 and DHA) in young adults. These results were largely consistent in direction (though smaller in magnitude and weaker in strength of evidence) in a two-sample MR analysis in an independent cohort of middle-aged adults. Results were attenuated, but consistent in direction, when potentially pleiotropic SNPs in the *FADS* gene regions were excluded. However, it should be noted that use of a narrow window for exclusion based on being within one of the three *FADS* genes may mean that some pleiotropic SNPs remain. Our subsequent forward MR analysis highlighted PUFAs as potentially having a causal role in the development of CRC.

Our analyses highlight a potentially important role of PUFAs in CRC liability. However, these analyses may be biased by substantial genetic pleiotropy among fatty acid traits. SNPs which are associated with levels of one fatty acid are generally associated with levels of many more fatty acid (and non-fatty acid) traits (*Borges et al., 2022b*). For instance, genetic instruments within the FADS cluster of genes will likely affect both omega-3 and omega-6 fatty acids, given *FADS1* and *FADS2* encode enzymes which catalyze the conversion of both from shorter chain into longer chain fatty acids (*Borges et al., 2022b*). In addition, the NMR metabolomics platform utilized in the analyses outlined here has limited coverage of fatty acids, meaning that many putative causal metabolites for CRC, for example, arachidonic acid, could not be investigated. Therefore, although our results indicate that PUFAs may be important in CRC risk, given the pleiotropic nature of the fatty acid genetic instruments and the limited coverage of the NMR platform, we are unable to determine with any certainty which specific classes of fatty acids may be driving these associations.

Our analyses featured evaluating the effect of genetic liability to CRC on circulating metabolites across repeated measures in the ALSPAC cohort. The mean ages at the time of the repeated measures were 8 y, 16 y, 18 y, and 25 y, representing childhood, early adolescence, late adolescence, and young adulthood, respectively, and therefore individuals in this cohort are unlikely to be taking metabolite-altering medication such as statins, and unlikely to have CRC. The strongest evidence for an effect of liability to CRC on metabolite levels was seen in late adolescence. The reason for this remains unclear. It is possible that this represents a true biological phenomenon if late adolescence is a critical window in CRC development or metabolite variability, which may be likely given the limited variance in metabolite levels at the later age of 25 y (*Supplementary file 1b*). The lack of an effect at the younger ages could be explained by the fact that the CRC GRS may capture many key life events or experiences which could impact the metabolome (e.g. initiation of smoking, higher category of BMI

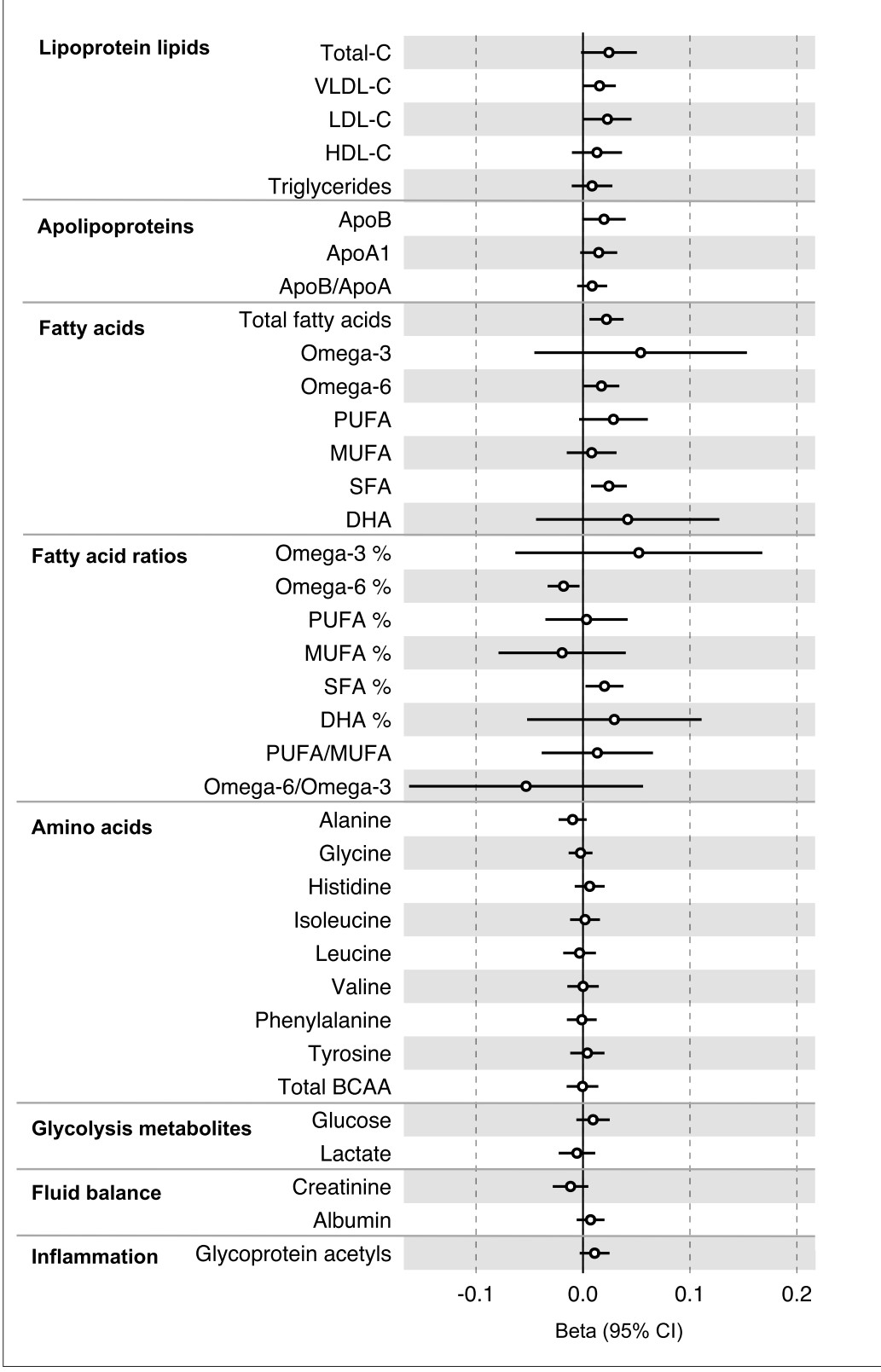

**Figure 3.** Associations of genetic liability to colorectal cancer with clinically validated metabolic traits in an independent sample of adults (UK Biobank, N = 118,466, median age 58 y) based on reverse two-sample Mendelian randomization analyses. Estimates shown are beta coefficients representing the SD-unit difference in

*Figure 3 continued on next page*

*Figure 3 continued*

metabolic trait per doubling of liability to colorectal cancer. Filled point estimates are those that pass a Benjamini–Hochberg FDR multiple-testing correction (FDR < 0.05).

The online version of this article includes the following figure supplement(s) for figure 3:

**Figure supplement 1.** Associations of genetic liability to colorectal cancer with clinically validated metabolic traits in an independent sample of adults based on reverse two-sample Mendelian randomization analyses.

**Figure supplement 2.** Associations of genetic liability to colorectal cancer (excluding genetic variants in the FADS gene region) with clinically validated metabolic traits in an independent sample of adults based on reverse two-sample Mendelian randomization analyses.

**Figure supplement 3.** Associations of genetic liability to colorectal and colon cancer with clinically validated metabolic traits in an independent sample of adults based on reverse two-sample Mendelian randomization analyses with FADS variants excluded from colorectal cancer instruments.

reached, educational attainment level set, etc.) but may not have yet happened at younger ages, thus obscuring an effect of genetic liability to CRC on the metabolome. Our results suggest that puberty could be important, with an effect seen seemingly particularly at the end of puberty. Repeating our analysis with sex-stratified data may aid in determining whether this is likely to be the case; sex-stratified GWAS for metabolites are not currently available to replicate such analyses. An alternative explanation is selection bias due to loss of follow-up, leading to a change in sample characteristics over time.

Another key finding in the reverse MR analysis was that genetic liability to CRC was associated with increased levels of total cholesterol, VLDL-cholesterol, LDL-cholesterol, and apolipoprotein B, though we find little evidence for a causal effect of these traits on risk of CRC in the forward MR, replicating previous forward MR analyses for total and LDL-cholesterol (*Gui et al., 2023*; *Rodriguez-Broadbent et al., 2017*; *Cornish et al., 2020*; *Luo et al., 2021*). This suggests that these traits may either be only predictive of (i.e. non-causal for) later CRC development or may be influenced by the development of CRC and could have diagnostic or predictive potential. Given that the participants in the ALSPAC cohort are many decades younger than the average age of diagnosis for CRC (mean age 25 y in the latest repeated measure analysed in ALSPAC; whereas the median age at diagnosis of CRC is 64 y) (*Granados-Romero et al., 2017*), the former seems the most likely scenario. Previous conventional observational studies have presented conflicting results when investigating the association between measures of cholesterol and CRC risk, with some finding an inverse association and others a positive association, possibly reflecting residual confounding in conventional observational analyses (*Park et al., 2000*; *Yang et al., 2022*; *Tian et al., 2015*; *Yao and Tian, 2015*; *Mayengbam et al., 2021*; *Mannes et al., 1986*; *Mamtani et al., 2016*). Previous MR studies have had similar findings to our forward MR analysis, in that there seems to be little evidence for a causal effect of cholesterol on CRC development (*Rodriguez-Broadbent et al., 2017*; *Cornish et al., 2020*; *Luo et al., 2021*). One possible explanation for how circulating levels of total cholesterol, VLDL-cholesterol, LDL-cholesterol, and apolipoprotein B could predict (without necessarily causing) future CRC development could be linked to diet. A previous MR analysis suggested an effect of increased BMI on several measures of circulating cholesterol (*Gui et al., 2023*). Consuming a diet which is high in fat may increase CRC risk both through and possibly independently of adiposity, alongside increasing levels of circulating cholesterol (*Ocvirk et al., 2019*; *Tang et al., 2012*; *Bell and Culp, 2022*; *Chiu et al., 2017*; *Han et al., 2018*; *Vincent et al., 2019*). The potential for lipoprotein or apolipoprotein lipid measures in future CRC risk prediction should be further investigated.

Our analyses stratified by anatomical subsite highlighted fatty acids as being affected by genetic liability to colon and proximal colon cancer, with the forward MR confirming that fatty acid traits may be particularly important in the development of these subsites of CRC as well as distal colon cancer.

In our forward MR analyses, we were unable to replicate the findings of three previous MR studies which found evidence for a causal effect of circulating linoleic acid levels on CRC development in terms of strength of evidence, though the direction of the effect estimate was similar to previous studies (*May Wilson et al., 2017*; *Khankari et al., 2020*; *Liyanage et al., 2019*). This is surprising as all three previous analyses had a much smaller sample size than that included in our analysis (the largest had sample size of 24,748 for exposure vs 118,466 presently; and 11,016 cases and 13,732 controls for outcome vs 52,775 cases and 45,940 controls presently). Our analysis using updated

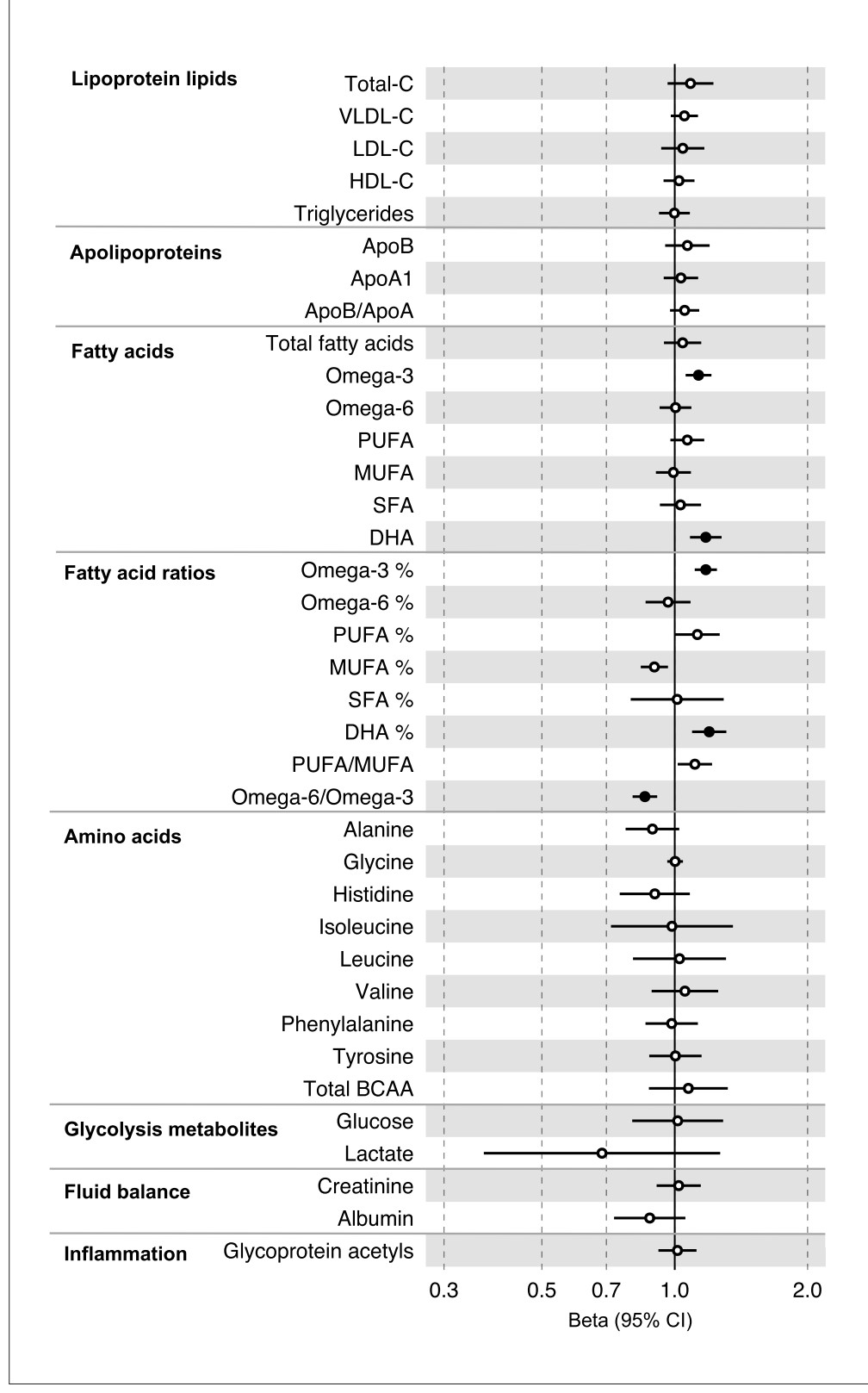

**Figure 4.** Associations of clinically validated metabolites with colorectal cancer based on conventional (forward) two-sample Mendelian randomization analyses in individuals from UK Biobank (N = 118,466, median age 58 y). Estimates shown are beta coefficients representing the logOR for colorectal cancer per SD metabolite. Filled point estimates are those that pass a Benjamini–Hochberg FDR multiple-testing correction (FDR < 0.05).

*Figure 4 continued on next page*

*Figure 4 continued*

The online version of this article includes the following figure supplement(s) for figure 4:

**Figure supplement 1.** Associations of clinically validated metabolites with colorectal cancer by site (colorectal, colon, distal colon, proximal colon, and rectal cancer) based on conventional (forward) two-sample Mendelian randomization analyses.

**Figure supplement 2.** Associations of clinically validated metabolites with colorectal cancer based on conventional (forward) two sample Mendelian randomization analyses with FADS variants excluded from metabolite instruments.

**Figure supplement 3.** Associations of clinically validated metabolites with colorectal cancer by site (colorectal, colon) based on conventional (forward) two sample Mendelian randomization analyses with FADS variants excluded from metabolite instruments.

---

genetic instruments to proxy fatty acids may be more successful in accurately instrumenting heterogenous phenotypes such as metabolite levels compared with previous analyses. All other findings in our forward MR analysis are consistent with previous MR studies where they exist (*Rodriguez-Broadbent et al., 2017*; *Cornish et al., 2020*; *Luo et al., 2021*).

## Limitations

The limitations of this study include firstly the relatively small sample size included in the ALSPAC analysis, which may have implications for power and precision. Secondly, mostly due to the longitudinal nature of the ALSPAC study, our sample at each time point is composed of slightly different individuals. This could be influencing our results and should be taken into account when comparing across time points. Thirdly, our analyses involving genetic instruments for CRC liability may have suffered from horizontal pleiotropy, even after excluding genetic variants in or near the *FADS* gene. Fourthly, our analyses were mostly restricted to white Europeans, which limits the generalizability of our findings to other populations. Fifthly, our analysis would benefit from being repeated with sex-stratified data. Although such GWAS results for metabolites are not currently available, the data to perform such GWAS are available in UK Biobank for future analyses. Sixthly, for our forward MR analysis, we used the UK Biobank for our exposure data. The UK Biobank has a median age of 58 at the time these measurements were taken, meaning statin use may be widespread in this population, which could be attenuating our effect estimates. Future work could attempt to replicate our analysis in a population with lower prevalence of statins intake. Finally, we included only metabolites measured using NMR. Confirming whether our results replicate using metabolite data measured with an alternative method would strengthen our findings.

## Conclusions

Our analysis provides evidence that genetic liability to CRC is associated with altered levels of metabolites at certain ages, some of which may have a causal role in CRC development. Further investigating the role of PUFAs in CRC risk and circulating cholesterol in CRC prediction may be promising avenues for future research.

## Acknowledgements

ALSPAC: We are extremely grateful to all the families who took part in this study, the midwives for their help in recruiting them, and the whole ALSPAC team, which includes interviewers, computer and laboratory technicians, clerical workers, research scientists, volunteers, managers, receptionists, and nurses. EPICOLON: We are sincerely grateful to all patients participating in this study who were recruited as part of the EPICOLON project. We acknowledge the Spanish National DNA Bank, Biobank of Hospital Clínic–IDIBAPS, and Biobanco Vasco for the availability of the samples. The work was carried out (in part) at the Esther Koplowitz Centre, Barcelona. JAB is supported by the Elizabeth Blackwell Institute for Health Research, University of Bristol, and the Wellcome Trust Institutional Strategic Support Fund (204813/Z/16/Z) and works in a Unit funded by the Medical Research Council (MC_UU_00011/1) and the University of Bristol. EEV, DNL, and CB are supported by Diabetes UK (17/0005587). NJT is a Wellcome Trust Investigator (202802/Z/16/Z), is the PI of the Avon Longitudinal Study of Parents and Children (MRC & WT 102215/2/13/2), is supported by the University of Bristol

NIHR Biomedical Research Centre (BRC-1215-20011), the MRC Integrative Epidemiology Unit (MC_UU_00011/1), and works within the CRUK Integrative Cancer Epidemiology Programme (C18281/A19169). EH is supported by a Cancer Research UK Population Research Committee Studentship (C18281/A30905), is supported by the CRUK Integrative Cancer Epidemiology Programme (C18281/A29019), and is part of the Medical Research Council Integrative Epidemiology Unit at the University of Bristol which is supported by the Medical Research Council (MC_UU_00011/4) and the University of Bristol. AC acknowledges funding from grant MR/N0137941/1 for the GW4 BIOMED MRC DTP, awarded to the Universities of Bath, Bristol, Cardiff, and Exeter from the Medical Research Council (MRC)/UKRI. MCB is supported by the UK Medical Research Council (MRC) Skills Development Fellowship (MR/P014054/1), University of Bristol Vice-Chancellor's Fellowship, and MRC Integrative Epidemiology Unit (MC_UU_00011/6). This work used the computational facilities of the Advanced Computing Research Centre, University of Bristol: http://www.bristol.ac.uk/acrc/. For study funding, please see Appendix 1.

## Additional information

### Funding

| Funder | Grant reference number | Author |
|---|---|---|
| Wellcome Trust | 10.35802/204813 | Joshua A Bell |
| Medical Research Council | MC_UU_00011/4 | Caroline Bull<br>Emma Hazelwood<br>Joshua A Bell<br>Vanessa Tan<br>Andrei-Emil Constantinescu<br>Carolina Borges<br>Nicholas J Timpson<br>Emma E Vincent |
| Diabetes UK | 17/0005587 | Caroline Bull<br>Danny Legge<br>Emma E Vincent |
| Wellcome Trust | 10.35802/202802 | Nicholas J Timpson |
| Wellcome Trust | 102215/2/13/2 | Nicholas J Timpson |
| Medical Research Council | 102215/2/13/2 | Nicholas J Timpson |
| Cancer Research UK | C18281/A19169 | Nicholas J Timpson |
| Cancer Research UK | C18281/A30905 | Emma Hazelwood |
| Cancer Research UK | C18281/A29019 | Emma Hazelwood |
| Medical Research Council | MR/N0137941/1 | Andrei-Emil Constantinescu |
| Medical Research Council | MR/P014054/1 | Carolina Borges |

The funders had no role in study design, data collection and interpretation, or the decision to submit the work for publication. For the purpose of Open Access, the authors have applied a CC BY public copyright license to any Author Accepted Manuscript version arising from this submission.

### Author contributions

Caroline Bull, Conceptualization, Data curation, Formal analysis, Visualization, Writing – original draft, Writing – review and editing; Emma Hazelwood, Formal analysis, Visualization, Methodology, Writing – original draft, Writing – review and editing; Joshua A Bell, Formal analysis, Methodology, Writing – review and editing; Vanessa Tan, Andrei-Emil Constantinescu, Writing – review and editing; Carolina Borges, Neil Murphy, Marc J Gunter, Methodology, Writing – review and editing; Danny Legge, Formal analysis, Writing – review and editing; Kimberley Burrows, Jeroen R Huyghe, Hermann Brenner, Sergi Castellvi-Bel, Andrew T Chan, Sun-Seog Kweon, Loic Le Marchand, Li Li, Iona Cheng, Rish K Pai,

Jane C Figueiredo, Data curation, Writing – review and editing; Nicholas J Timpson, Funding acquisition, Methodology, Writing – review and editing; Emma E Vincent, Funding acquisition, Methodology, Writing – original draft, Writing – review and editing

## Author ORCIDs

Caroline Bull ⓘ https://orcid.org/0000-0002-2176-5120
Emma Hazelwood ⓘ http://orcid.org/0000-0002-4888-6037
Vanessa Tan ⓘ http://orcid.org/0000-0001-7938-127X
Danny Legge ⓘ http://orcid.org/0000-0002-3897-5861
Jeroen R Huyghe ⓘ https://orcid.org/0000-0001-6027-9806
Emma E Vincent ⓘ http://orcid.org/0000-0002-8917-7384

## Ethics

Written informed consent was obtained for all study participants. Ethical approval was obtained from the ALSPAC Law and Ethics Committee and the local research ethics committee (proposal B3538). Consent for biological samples has been collected in accordance with the Human Tissue Act (2004). Informed consent for the use of data collected via questionnaires and clinics was obtained from participants following the recommendations of the ALSPAC Ethics and Law Committee at the time. Ethics for the CRC GWAS were approved by respective institutional review boards.

Joint Public Review: https://doi.org/10.7554/eLife.87894.3.sa1
Author Response https://doi.org/10.7554/eLife.87894.3.sa2

---

# Additional files

## Supplementary files

• Supplementary file 1. Supplementary tables. (**a**) Genetic variants used to construct genetic risk scores reflecting colorectal cancer liability. (**b**) Mean and SD values for raw metabolic traits at different life stages among ALSPAC offspring. (**c**) Associations of genetic liability to colorectal cancer with metabolic traits at different early life stages among ALSPAC offspring. (**d**) Associations of genetic liability to colorectal cancer (excluding SNPs in the FADS gene region) with metabolic traits at different early life stages among ALSPAC offspring. (**e**) Assessment of instrument strength for MR analyses. (**f**) Associations of genetic liability to colorectal cancer with metabolic traits based on two-sample MR. (**g**) Associations of genetic liability to colorectal cancer with metabolic traits based on two-sample MR, excluding variants in the FADS gene region. (**h**) Genetic variants used to instrument circulating metabolites. (**i**) Assessment of instrument strength for MR analyses. (**j**) Estimated effects of circulating metabolites on colorectal cancer risk based on two-sample MR. (**k**) Estimated effects of circulating metabolites on colorectal cancer risk based on two-sample MR, excluding variants in the FADS gene region.

• MDAR checklist

## Data availability

All data generated during this study are included in the manuscript and supporting file. Access to individual-level ALSPAC data are available following an application. The summary-level GWAS data for CRC used in this study are available following an application to GECCO (managed access). All data generated by this study are available in the manuscript and supporting material. R scripts used in this study have been made publicly available on GitHub at: https://github.com/cb12104/adiposity_metabolites_crc, (copy archived at *Bull, 2020*).

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

## Appendix 1

### Study funding

#### ALSPAC

The UK Medical Research Council and Wellcome (grant ref: 217065/Z/19/Z) and the University of Bristol provide core support for ALSPAC. This publication is the work of the authors who will serve as guarantors for theon the ALSPAC websiteon the ALSPAC websiteon the ALSPAC website contents of this paper. A comprehensive list of grants funding is available on the ALSPAC website (http://www.bristol.ac.uk/alspac/external/documents/grant-acknowledgements.pdf); this research was specifically funded by the UK Medical Research Council (grant ref: MC_UU_12013/1). GWAS data was generated by Sample Logistics and Genotyping Facilities at Wellcome Sanger Institute and LabCorp (Laboratory Corporation of America) using support from 23andMe.

#### Genetics and Epidemiology of Colorectal Cancer Consortium (GECCO)

National Cancer Institute, National Institutes of Health, U.S. Department of Health and Human Services (U01 CA137088, R01 CA059045, R01 201407). Genotyping/Sequencing services were provided by the Center for Inherited Disease Research (CIDR) contract number HHSN268201700006I and HHSN268201200008I. This research was funded in part through the NIH/NCI Cancer Center Support Grant P30 CA015704. Scientific Computing Infrastructure at Fred Hutch funded by ORIP grant S10OD028685.

#### ASTERISK

A Hospital Clinical Research Program (PHRC-BRD09/C) from the University Hospital Center of Nantes (CHU de Nantes) and supported by the Regional Council of Pays de la Loire, the Groupement des Entreprises Françaises dans la Lutte contre le Cancer (GEFLUC), the Association Anne de Bretagne Génétique, and the Ligue Régionale Contre le Cancer (LRCC).

The ATBC Study is supported by the Intramural Research Program of the U.S. National Cancer Institute, National Institutes of Health, Department of Health and Human Services.

CLUE II funding was from the National Cancer Institute (U01 CA086308, Early Detection Research Network; P30 CA006973), National Institute on Aging (U01 AG018033), and the American Institute for Cancer Research. The content of this publication does not necessarily reflect the views or policies of the Department of Health and Human Services, nor does mention of trade names, commercial products, or organizations imply endorsement by the US government.

#### Maryland Cancer Registry (MCR)

Cancer data was provided by the Maryland Cancer Registry, Center for Cancer Prevention and Control, Maryland Department of Health, with funding from the State of Maryland and the Maryland Cigarette Restitution Fund. The collection and availability of cancer registry data are also supported by the Cooperative Agreement NU58DP006333, funded by the Centers for Disease Control and Prevention. Its contents are solely the responsibility of the authors and do not necessarily represent the official views of the Centers for Disease Control and Prevention or the Department of Health and Human Services.

#### ColoCare

This work was supported by the National Institutes of Health (grant numbers R01 CA189184 [Li/Ulrich], U01 CA206110 [Ulrich/Li/Siegel/Figueiredo/Colditz, 2P30CA015704-40; Gilliland, R01 CA207371 Ulrich/Li]), the Matthias Lackas-Foundation, the German Consortium for Translational Cancer Research, and the EU TRANSCAN initiative.

The Colon Cancer Family Registry (CCFR, https://www.coloncfr.org/) is supported in part by funding from the National Cancer Institute (NCI), National Institutes of Health (NIH) (award U01 CA167551). Support for case ascertainment was provided in part from the Surveillance, Epidemiology, and End Results (SEER) Program and the following US state cancer registries: AZ, CO, MN, NC, NH; and by the Victoria Cancer Registry (Australia) and Ontario Cancer Registry (Canada). The CCFR Set-1 (Illumina 1M/1M-Duo) and Set-2 (Illumina Omni1-Quad) scans were supported by NIH awards U01 CA122839 and R01 CA143237 (to GC). The CCFR Set-3 (Affymetrix Axiom CORECT Set array) was supported by NIH award U19 CA148107 and R01 CA81488 (to SBG). The CCFR Set-4 (Illumina

OncoArray 600K SNP array) was supported by NIH award U19 CA148107 (to SBG) and by the Center for Inherited Disease Research (CIDR), which is funded by the NIH to the Johns Hopkins University, contract number HHSN268201200008I. Additional funding for the OFCCR/ARCTIC was through award GL201-043 from the Ontario Research Fund (to BWZ), award 112746 from the Canadian Institutes of Health Research (to TJH), through a Cancer Risk Evaluation (CaRE) Program grant from the Canadian Cancer Society (to SG), and through generous support from the Ontario Ministry of Research and Innovation. The SFCCR Illumina HumanCytoSNP array was supported in part through NCI/NIH awards U01/U24 CA074794 and R01 CA076366 (to PAN). The content of this manuscript does not necessarily reflect the views or policies of the NCI, NIH or any of the collaborating centres in the Colon Cancer Family Registry (CCFR), nor does mention of trade names, commercial products, or organizations imply endorsement by the US Government, any cancer registry, or the CCFR.

## COLON
The COLON study is sponsored by Wereld Kanker Onderzoek Fonds, including funds from grant 2014/1179 as part of the World Cancer Research Fund International Regular Grant Programme, by Alpe d'Huzes and the Dutch Cancer Society (UM 2012-5653, UW 2013-5927, UW2015-7946), and by TRANSCAN (JTC2012-MetaboCCC, JTC2013-FOCUS). The Nqplus study is sponsored by a ZonMW investment grant (98-10030); by PREVIEW, the project PREVention of diabetes through lifestyle intervention and population studies in Europe and around the World (PREVIEW) project which received funding from the European Union Seventh Framework Programme (FP7/2007–2013) under grant no. 312057; by funds from TI Food and Nutrition (cardiovascular health theme), a public–private partnership on precompetitive research in food and nutrition; and by FOODBALL, the Food Biomarker Alliance, a project from JPI Healthy Diet for a Healthy Life.

## COLO2&3
National Institutes of Health (R01 CA060987).

## Colorectal Cancer Transdisciplinary (CORECT) study
The CORECT study was supported by the National Cancer Institute, National Institutes of Health (NCI/NIH), U.S. Department of Health and Human Services (grant numbers U19 CA148107, R01 CA081488, P30 CA014089, R01 CA197350; P01 CA196569; R01 CA201407; R01 CA242218), National Institutes of Environmental Health Sciences, National Institutes of Health (grant number T32 ES013678), and a generous gift from Daniel and Maryann Fong.

## CORSA
The CORSA study was funded by the Austrian Research Funding Agency (FFG) BRIDGE (grant 829675, to Andrea Gsur), the 'Herzfelder'sche Familienstiftung' (grant to Andrea Gsur) and was supported by COST Action BM1206.

## CPS-II
The American Cancer Society funds the creation, maintenance, and updating of the Cancer Prevention Study-II (CPS-II) cohort. The study protocol was approved by the institutional review boards of Emory University, and those of participating registries as required.

## CRCGEN
The Colorectal Cancer Genetics & Genomics, Spanish study was supported by Instituto de Salud Carlos III, co-funded by FEDER funds – a way to build Europe (grants PI14-613 and PI09-1286), Agency for Management of University and Research Grants (AGAUR) of the Catalan Government (grant 2017SGR723), Junta de Castilla y León (grant LE22A10-2), the Spanish Association Against Cancer (AECC) Scientific Foundation grant GCTRA18022MORE, and the Consortium for Biomedical Research in Epidemiology and Public Health (CIBERESP), action Genrisk. Sample collection of this work was supported by the Xarxa de Bancs de Tumors de Catalunya sponsored by Pla Director d'Oncología de Catalunya (XBTC), Plataforma Biobancos PT13/0010/0013, and ICOBIOBANC, sponsored by the Catalan Institute of Oncology. We thank CERCA Programme, Generalitat de Catalunya, for institutional support.

## Czech Republic CCS
This work was supported by the Grant Agency of the Czech Republic (21-04607X, 20-03997S), the Grant Agency of the Ministry of Health of the Czech Republic (grants AZV NU21-07-00247 and AZV NU21-03-00506), and Charles University Research Fund (Cooperation 43-Surgical disciplines)

## DACHS
This work was supported by the German Research Council (BR 1704/6-1, BR 1704/6-3, BR 1704/6-4, CH 117/1-1, HO 5117/2-1, HE 5998/2-1, KL 2354/3-1, RO 2270/8-1, and BR 1704/17-1), the Interdisciplinary Research Program of the National Center for Tumor Diseases (NCT), Germany, and the German Federal Ministry of Education and Research (01KH0404, 01ER0814, 01ER0815, 01ER1505A, and 01ER1505B).

## DALS
National Institutes of Health (R01 CA048998 to ML Slattery).

## EDRN
This work is funded and supported by the NCI, EDRN Grant (U01-CA152753).

## EPIC
The coordination of EPIC is financially supported by the International Agency for Research on Cancer (IARC) and also by the Department of Epidemiology and Biostatistics, School of Public Health, Imperial College London, which has additional infrastructure support provided by the NIHR Imperial Biomedical Research Centre (BRC). The national cohorts are supported by Danish Cancer Society (Denmark); Ligue Contre le Cancer, Institut Gustave Roussy, Mutuelle Générale de l'Education Nationale, Institut National de la Santé et de la Recherche Médicale (INSERM) (France); German Cancer Aid, German Cancer Research Center (DKFZ), German Institute of Human Nutrition Potsdam- Rehbruecke (DIfE), Federal Ministry of Education and Research (BMBF) (Germany); Associazione Italiana per la Ricerca sul Cancro-AIRC-Italy, Compagnia di SanPaolo and National Research Council (Italy); Dutch Ministry of Public Health, Welfare and Sports (VWS), Netherlands Cancer Registry (NKR), LK Research Funds, Dutch Prevention Funds, Dutch ZON (Zorg Onderzoek Nederland), World Cancer Research Fund (WCRF), Statistics Netherlands (The Netherlands); Health Research Fund (FIS) – Instituto de Salud Carlos III (ISCIII), Regional Governments of Andalucía, Asturias, Basque Country, Murcia and Navarra, and the Catalan Institute of Oncology – ICO (Spain); Swedish Cancer Society, Swedish Research Council and Region Skåne and Region Västerbotten (Sweden); Cancer Research UK (14136 to EPIC-Norfolk; C8221/A29017 to EPIC-Oxford), Medical Research Council (1000143 to EPIC-Norfolk; MR/M012190/1 to EPIC-Oxford) (the United Kingdom).

## EPICOLON
This work was supported by grants from Fondo de Investigación Sanitaria/FEDER (PI08/0024, PI08/1276, PS09/02368, PI11/00219, PI11/00681, PI14/00173, PI14/00230, PI17/00509, 17/00878, PI20/00113, PI20/00226, Acción Transversal de Cáncer), Xunta de Galicia (PGIDIT07PXIB9101209PR), Ministerio de Economia y Competitividad (SAF07-64873, SAF 2010-19273, SAF2014-54453R), Fundación Científica de la Asociación Española contra el Cáncer (GCB13131592CAST, PRYGN211085CAST), Beca Grupo de Trabajo 'Oncología' AEG (Asociación Española de Gastroenterología), Fundación Privada Olga Torres, FP7 CHIBCHA Consortium, Agència de Gestió d'Ajuts Universitaris i de Recerca (AGAUR, Generalitat de Catalunya, 2014SGR135, 2014SGR255, 2017SGR21, 2017SGR653, 2021SGR00716, 2021SGR01185), Catalan Tumour Bank Network (Pla Director d'Oncologia, Generalitat de Catalunya), PERIS (SLT002/16/00398, Generalitat de Catalunya), Marató TV3 (202008-10), CERCA Programme (Generalitat de Catalunya), and COST Actions BM1206 and CA17118. CIBERehd is funded by the Instituto de Salud Carlos III.ESTHER/VERDI. This work was supported by grants from the Baden-Württemberg Ministry of Science, Research and Arts, and the German Cancer Aid.

## Harvard cohorts
HPFS is supported by the National Institutes of Health (P01 CA055075, UM1 CA167552, U01 CA167552, R01 CA137178, R01 CA151993, and R35 CA197735), NHS by the National Institutes of

Health (P01 CA087969, UM1 CA186107, R01 CA137178, R01 CA151993, and R35 CA197735), and PHS by the National Institutes of Health (R01 CA042182).

### Hawaii Adenoma Study
NCI grants R01 CA072520.

### HCES-CRC
The Hwasun Cancer Epidemiology Study – Colon and Rectum Cancer (HCES-CRC; grants from Chonnam National University Hwasun Hospital, HCRI15011-1).

### Kentucky
This work was supported by the following grant support: Clinical Investigator Award from Damon Runyon Cancer Research Foundation (CI-8); NCI R01CA136726.

### LCCS
The Leeds Colorectal Cancer Study was funded by the Food Standards Agency and Cancer Research UK Programme Award (C588/A19167).

MCCS cohort recruitment was funded by VicHealth and Cancer Council Victoria. The MCCS was further supported by Australian NHMRC grants 509348, 209057, 251553, and 504711 and by infrastructure provided by Cancer Council Victoria. Cases and their vital status were ascertained through the Victorian Cancer Registry (VCR) and the Australian Institute of Health and Welfare (AIHW), including the National Death Index and the Australian Cancer Database. BMLynch was supported by MCRF18005 from the Victorian Cancer Agency.

### MEC
National Institutes of Health (R37 CA054281, P01 CA033619, and R01 CA063464).

### MECC
This work was supported by the National Institutes of Health, U.S. Department of Health and Human Services R01 CA081488, R01 CA197350, U19 CA148107, R01 CA242218, and a generous gift from Daniel and Maryann Fong.

### MSKCC
The work at Sloan Kettering in New York was supported by the Robert and Kate Niehaus Center for Inherited Cancer Genomics and the Romeo Milio Foundation. Moffitt: This work was supported by funding from the National Institutes of Health (grant numbers R01 CA189184, P30 CA076292), Florida Department of Health Bankhead-Coley Grant 09BN-13, and the University of South Florida Oehler Foundation. Moffitt contributions were supported in part by the Total Cancer Care Initiative, Collaborative Data Services Core, and Tissue Core at the H. Lee Moffitt Cancer Center & Research Institute, a National Cancer Institute-designated Comprehensive Cancer Center (grant number P30 CA076292).

### NCCCS I and II
We acknowledge funding support for this project from the National Institutes of Health, R01 CA066635 and P30 DK034987.

### NFCCR
This work was supported by an Interdisciplinary Health Research Team award from the Canadian Institutes of Health Research (CRT 43821); the National Institutes of Health, U.S. Department of Health and Human Services (U01 CA074783); and National Cancer Institute of Canada grants (18223 and 18226). The authors wish to acknowledge the contribution of Alexandre Belisle and the genotyping team of the McGill University and Génome Québec Innovation Centre, Montréal, Canada, for genotyping the Sequenom panel in the NFCCR samples. Funding was provided to Michael O. Woods by the Canadian Cancer Society Research Institute.

## NSHDS
The research was supported by Biobank Sweden through funding from the Swedish Research Council (VR 2017-00650, VR 2017-01737), the Swedish Cancer Society (CAN 2017/581), Region Västerbotten (VLL-841671, VLL-833291), Knut and Alice Wallenberg Foundation (VLL-765961), and the Lion's Cancer Research Foundation (several grants) and Insamlingsstiftelsen, both at Umeå University.

## OSUMC
OCCPI funding was provided by Pelotonia and HNPCC funding was provided by the NCI (CA016058 and CA067941).

## PLCO
Intramural Research Program of the Division of Cancer Epidemiology and Genetics and supported by contracts from the Division of Cancer Prevention, National Cancer Institute, NIH, DHHS. Funding was provided by National Institutes of Health (NIH), Genes, Environment and Health Initiative (GEI) Z01 CP 010200, NIH U01 HG004446, and NIH GEI U01 HG 004438.

## SEARCH
The University of Cambridge has received salary support in respect of PDPP from the NHS in the East of England through the Clinical Academic Reserve. Cancer Research UK (C490/A16561); the UK National Institute for Health Research Biomedical Research Centres at the University of Cambridge.

## SELECT
Research reported in this publication was supported in part by the National Cancer Institute of the National Institutes of Health under Award Numbers U10 CA037429 (CD Blanke), and UM1 CA182883 (CM Tangen/IM Thompson). The content is solely the responsibility of the authors and does not necessarily represent the official views of the National Institutes of Health.

## SMS and REACHS
This work was supported by the National Cancer Institute (grant P01 CA074184 to JDP and PAN, grants R01 CA097325, R03 CA153323, and K05 CA152715 to PAN), and the National Center for Advancing Translational Sciences at the National Institutes of Health (grant KL2 TR000421 to ANB-H).

## The Swedish Low-Risk Colorectal Cancer Study
The study was supported by grants from the Swedish research council; K2015-55X-22674-01-4, K2008-55X-20157-03-3, K2006-72X-20157-01-2, and the Stockholm County Council (ALF project).

## Swedish Mammography Cohort and Cohort of Swedish Men
This work is supported by the Swedish Research Council /Infrastructure grant, the Swedish Cancer Foundation, and the Karolinska Institute's Distinguished Professor Award to Alicja Wolk.

## UK Biobank
This research has been conducted using the UK Biobank Resource under Application Number 8614

## VITAL
National Institutes of Health (K05 CA154337).

The WHI programme is funded by the National Heart, Lung, and Blood Institute, National Institutes of Health, U.S. Department of Health and Human Services through contracts 75N92021D00001, 75N92021D00002, 75N92021D00003, 75N92021D00004, and 75N92021D00005.

