## [Editor Report · eLife assessment]

The manuscript by Bull et al. provides **valuable** information on the relationship between metabolic features, in particular different lipoproteins and fatty acids, and colorectal cancer. They use **solid** methods and combine different data sources to analyze forward and reverse Mendelian randomizations that support their claims.

---

## [Referee Report · Joint Public Review]

Bull et al aimed to use data from observational studies and mendelian randomisation to explore if changes in circulating metabolites are associated with colorectal cancer development. As Mendelian randomisation uses information on genetic variations which are fixed at birth, it is less vulnerable to confounding than standard observational studies.

Overall, a major strength of the study is that it uses data from large cohort studies, one from childhood, adolescence, and early adulthood when the incidence of colorectal cancer is very low (reducing the likelihood of reverse causation) and before medication (such as statins which have the potential to affect metabolite levels) has been initiated.

This study has some weaknesses which have been acknowledged by the authors. Although the findings of this study indicate the potentially significant role that polyunsaturated fatty acids may have in colorectal cancer risk, the genes and therefore also the genetic variations (SNPs) associated with fatty acids often produce an effect for more than one fatty acid which may introduce bias. This together with the fact that there was limited information available on many specific fatty acids which are known causative metabolites for colorectal cancer, makes it difficult to establish with confidence which specific classes of fatty acids could potentially play a causative role in these associations. Also, the study populations are majority white European descent which may limit the generalizability of these findings to other populations.

The methodology used was largely acceptable to achieve the aims set out and the findings have shown an association between polyunsaturated fat levels and genetic liability to colorectal cancer.

Overall, this is an important piece of work which has the potential to contribute to our understanding of the causal relationship between circulating metabolites at different stages of the life cycle and colorectal cancer risk as it would be extremely difficult to gather such evidence using other study designs. It opens the door for future research aiming to better understand the role that these metabolites could play in colorectal cancer risk prediction and in turn help identify groups of individuals who would benefit most from prevention and early detection interventions.

This work will be of interest not only to epidemiologists working in the area of GI tract cancers but also those interested in the different applications for mendelian randomisation within cancer epidemiology research.

---

## [Author Response]

The following is the authors’ response to the original reviews.

We would like to thank the reviewers for their helpful comments which we have addressed, point-by-point, below:

**Reviewer #1:**
1. It might be useful to add more details to the methods (especially lines 191-196) to make them a bit more user-friendly for an audience who still may be unfamiliar with the relatively new and complex Mendelian randomisation technique.

The following information has been included in this section of the methods, to describe the different MR models in more detail:

“The IVW MR model will produce biased effect estimates in the presence of horizontal pleiotropy, i.e. where one or more genetic variant(s) included in the instrument affect the outcome by a pathway other than through the exposure. In the weighted median model, each genetic variant is weighted according to its distance from the median effect of all genetic variants. Thus, the weighted median model will provide an unbiased estimate when at least 50% of the information in an instrument comes from genetic variants that are not horizontally pleiotropic. The weighted mode model uses a similar approach but weights genetic instruments according to the mean effect. In this model, over 50% of the weight of the genetic instrument can be contributed to by genetic variants which are horizontally pleiotropic, but the most common amount of pleiotropy must be zero (known as the Zero Modal Pleiotropy Assumption (ZEMPA))[Hartwig et al., 2017].”

2. I was just wondering why MR egger was not carried out as part of this analysis?

We did consider also employing the MR Egger model as a further sensitivity analysis. However, given we were already employing the weighted median and weighted mode models, and given that MR-Egger suffers from reduced statistical power in comparison to the other models, we reasoned that adding in a further MR model would not add further clarity to our analyses, particularly given the relatively small sample size.

3. Although it is included in Figure 1 flowchart, I think it is also important to explain clearly in the written text way only n=6,118 of n=13,988 children in ALSPAC study were included in this study and the reason for this.

The following information has been included in the paragraph describing the ALSPAC study in the methods:

“Sufficient information was available on 6,221 of these individuals to be included in our analysis, as metabolomics was not performed for all individuals in the ALSPAC study.”

4. It is mentioned within the discussion 'the NMR metabolomics platform utilised in the analyses outlined here has limited coverage of fatty acids'. I think it might be useful to also add this detail into the methods section to aid readers when they are making their own interpretation whilst reading the results section.

The following sentence has been included in the methods section:

“This metabolomics platform has limited coverage of fatty acids.”

5. However, I feel that the conclusion should be tempered slightly as although this study alongside other similar MR studies provides evidence of an association between genetic liability to CRC and levels of metabolites at certain ages, I do not think there is enough evidence at this stage to say that genetic liability for CRC actually alters the levels of metabolites.

The first sentence of the conclusion has been changed to:

“Our analysis provides evidence that genetic liability to CRC is associated with altered levels of metabolites at certain ages, some of which may have a causal role in CRC development.”

**Reviewer #2:**
1. The background is lacking introduction to the different components of the metabolic features tested. For instance, there is a broader discussion about polyunsaturated fatty acids (PUFA) in the discussion, however, this should have been introduced and defined already before that. What metabolites are included in that term (PUFA)? Are there other studies on PUFA and CRC?

The following information has been included in the background section:

“In particular, previous work has highlighted polyunsaturated fatty acids (PUFA) as potentially having a role in colorectal cancer development. The term PUFA includes omega-3 and -6 fatty acids. Recent MR work has highlighted a possible link between PUFAs, in particular omega 6 PUFAs, and colorectal cancer risk.”

2. There seem to be indications for horizontal pleiotropy given the changed estimates when genetic variants in the FADS loci are removed. Could multivariable MR methods have been used to account for pleiotropy and differentiate individual fatty acid effects?

Multivariable MR can be employed to investigate the effects of horizontal pleiotropy. However, the multiple exposures must have sufficiently distinct underlying genetic architecture in order to instrument each one whilst adjusting for the other, as determined by conditional F-statistics. Given the correlations across metabolite levels, this is unlikely to be the case.

3. The ALSPAC sample sizes are decreasing across the different age groups, which is not strange given the longitudinal collection. However, does the altered sample composition affect the results? Have sensitivity analyses been done on the complete set of individuals from age 8-25?

The altered sample composition could be affecting results. The limitations section of the discussion has been amended to reflect this:

“Secondly, mostly due to the longitudinal nature of the ASLAPC study, our sample at each time point is composed of slightly different individuals. This could be influencing our results, and should be taken into account when comparing across time points.”

We have not completed any sensitivity analyses to investigate this.

4. Although beyond the scope of this paper, sex-stratified GWAS analyses on metabolites can easily be done in UK Biobank.

We thank the reviewer for this suggestion, and agree that this would be an interesting future analysis. We have amended the discussion to mention this:

“Fourthly, our analysis would benefit from being repeated with sex-stratified data. Although such GWAS results for metabolites are not currently available, the data to perform such GWAS are available in UK Biobank for future analyses.”

5. Very minor, there is a difference in reporting a number of decimals in ALSPAC results.There is also a difference in reporting the units for the results comparing text and figures (per SD higher CRC liability or per doubling). Please include sample sizes and data sources in the figure legends as they should be stand-alone items.

We have amended the ALSPAC results to all have two decimal places, reporting units have been altered and figure legends to include sample sizes and data sources.